# OpenReview forum: "Mitigating Jailbreaks with Intent-Aware LLMs"
_ICLR.cc/2026/Conference — Submitted to ICLR 2026_

### Official Review · Reviewer_1TYP · 2025-10-16

**Soundness:** 3
**Presentation:** 3
**Contribution:** 2
**Rating:** 4
**Confidence:** 4

**Summary:**

This paper proposes \textsc{Intent-FT}, a simple and lightweight fine-tuning approach that explicitly trains large language models (LLMs) to infer the underlying intent of an instruction before generating responses. The authors conduct comprehensive evaluations on both parametric and non-parametric attacks across open-source and proprietary models, demonstrating that the method consistently mitigates all evaluated jailbreaking attacks.

**Strengths:**

1. The motivation and method are clear, and the visualization is helpful.

2. The authors provide sufficient evidence, including extensive experiments and ablation studies, to support the effectiveness of their proposed method.

3. This paper is well-written, making it easy to follow.

**Weaknesses:**

1. The relationship between intent perception and jailbreaking robustness has been extensively explored from both attack and defense perspectives. This paper appears to trivially transfer this perspective into a fine-tuning setting, raising concerns about its novelty and conceptual contribution.

2. What is the difference between <intent> inference and the reasoning-based model (e.g., models that “think before answering”), such as GPT-5? As the reasoning model is inherent to be able to better extract the intent and show better jailbreaking robustness.

3. In my opinion, white-box GCG and GCG-like attacks, which rely on gradient-based optimization, can almost always jailbreak a model given a sufficient attack budget. Therefore, it is unclear why the authors claim an ASR below 50% and do not provide further results.

**Questions:**

Refer to weaknesses

---

> ### Author Response · Authors · 2025-11-12
>
> Dear reviewer 1TYP,
>
> We are extremely thankful for your informative and interesting feedback. We were especially excited that you found our work to be clear and objective to be well-motivated. We are grateful for your acknowledgement on the comprehensive evaluation performed.
>
> We will address your remaining concerns below.
>
> **1. Intent perception and Jailbreaking robustness is already extensively explored**
>
> While intent perception and jailbreak robustness have been explored in prior works, our Intent-FT contributes new empirical insights that extend beyond surface-level similarities. Specifically, our experiments uncover the brittleness of prompting-based Intent Analysis (IA) — such approaches are easily subverted by more invasive jailbreaks compared to Harmful-FT and induce significant over-refusal and utility degradation. This is primarily due to IA’s redundant self-reminding behavior (e.g., prompting the model twice to stay safe), which over-regularizes the model toward excessive caution.
>
> Moreover, our results demonstrate that training solely on safety exemplars (Safety-FT) is insufficient against strong adaptive adversaries such as PAIR and Adversarial Alignment (AA). In contrast, incorporating a lightweight intention reasoning step before generation markedly improves robustness while preserving task utility.
>
> This thus strengths our claim that allocating a small budget to **train the model explicitly to reason about the intentions** would reduce over-refusal, incur minimal utility degradation and provide maximal defense against jailbreak attacks.
>
> 2. **Difference between [object Object] inference and reasoning-based model**
>
> We are not fully certain what does the reviewer mean by [object Object] inference. **However, we will make an educated guess that the reviewer is asking if inference-based defenses such as IA would have better results if implemented in reasoning-based models**
>
> We compare Intent-FT with IA on a reasoning model: DeepSeek-R1-Distill-Llama-8B. Note that here, we append the intent reasoning as the starting portion of the thinking process. The results on ASR and over-refusal is shown below.
>
> | Method        | PAIR | Harmful-FT | Prefill | Over-refusal |
> |:--------------|------:|------:|------:|------:|
> | IA     | 88  | 44  |  25      | 7.2 |
> | Intent-FT     | **18** | **4** | **14** | **3.6** |
>
> Our findings indicates that even thinking models which possess more better reasoning capabilities are still suspectible to jailbreak attacks. This may not be suprisingly since 1) thinking models are optimized for downstream task performance instead of safety, hence the added reasoning does not yield any positive delta over non-thinking models, 2) Past works such as [1] has shown that models optimized for performance introduces a negative effect on its safety training since the desire to complete a task contrasts the desire to refuse it.
>
> [1] Qi, Xiangyu, et al. "Fine-tuning Aligned Language Models Compromises Safety, Even When Users Do Not Intend To!", ICLR 2024
>
> 3. **GCG attacks should be successful**
>
> We did ran gradient optimization-based attacks such as GCG and AutoDAN on Llama-3.1. We followed the original implemented provided in the codebase of GCG and paper and ran each prompt for $500$ steps. However, we found that despite the high number of steps required, it only achieved an **ASR of 23%**. We attribute the reasons being that Llama-3.1 has undergone a more rigorous safety training reigme as compared to the older models (Llama-2) as tested in the original work. This essentially makes GCG an undesirable attacking strategy as compared to PAIR given that it 1) requires lesser steps and compute, 2) converges quicker and more effectively, 3) does not require gradients (**this is an important requirement to attack closed-source models**). Thus, we limited our study to the more effective optimization attacks (PAIR and AA) which has been very effectiveness when used with stronger attacker models like GPT-4.1. We also observe similar findings with AutoDAN, achieving an ASR of only **36%** while being similarly expensive as GCG.
>
> Once again, we thank you for taking the time to review our work and engaging in a interesting and helpful discussion of our work! We hope that the above comments have addressed your concerns, else we will be more than happy to provide further clarifications  that can encourage you to improve your score.

---

> ### Author Response · Authors · 2025-11-27
> **Kind reminders**
>
> Dear Reviewer 1TYP,
>
> We sincerely thank the reviewer once again for their careful reading and constructive feedback, which have helped us significantly improve the quality and clarity of this work. We have addressed all raised concerns in detail in the rebuttal and corresponding revisions. We kindly invite the reviewer to reassess the manuscript in light of these changes, and if the responses have satisfactorily resolved the concerns, we would greatly appreciate reconsideration of the scores accordingly.

---

### Official Review · Reviewer_ysq3 · 2025-10-26

**Soundness:** 3
**Presentation:** 2
**Contribution:** 3
**Rating:** 6
**Confidence:** 4

**Summary:**

This paper proposes a simple yet effective method for safety alignment, Intent-FT. Similar to IA (Zhang et al., 2024), Intent-FT encourages LLMs to infer and articulate the user's intent. The key difference is that it trains the model to generate intention CoTs enclosed between `<intent>` and `</intent>` tokens, using intention CoTs collected from a strong teacher model. Intent-FT demonstrates superior performance compared to relevant baselines in both safety alignment and utility.

**Strengths:**

- The motivation is intuitive, and the method is simple and broadly applicable.

- Although it introduces additional computational cost during fine-tuning, the overhead appears to be marginal (not sure), and the empirical results are strong.

**Weaknesses:**

This paper presents a simple yet effective method that is appealing to me, but I believe it can be improved with the following recommendations:

- Important experimental setup details are missing. For example, the fine-tuning appears marginal (100 samples, as noted in line 192), but key details such as the number of epochs/iterations, wall-clock fine-tuning time, and whether LoRA or full fine-tuning was used are not provided.

- Presentation could be improved:
  - The notation is confusing. Subscripts are sometimes used for indexing (e.g., $x_{1:T}$) and sometimes to indicate different instructions (e.g., $q_v, q_a$). Notations such as $q_v, q_a$ and $y_ρ^h, y_s, y_i$ are hard to distinguish.
  - In line 345, the authors claim that Intent-FT optimizes the trade-off along the Pareto frontier. It would be clearer to include a plot illustrating this claim.
  - Several important figures referenced in the main text (e.g., Figures 5, 7, 12) are placed in the Appendix. Although this may be due to page limits, including at least some of these figures in the main paper would help clarify the effectiveness of the method.

- While the proposed idea is simple and broadly applicable, the experimental evaluation is limited. Although Llama and Qwen are representative models, demonstrating results on a wider variety of backbone architectures (e.g., Gemma) would better support claims of generalizability.

- Intent-FT shows that a stronger teacher model can generate intention CoTs to train smaller LLMs and improve their safety alignment. This implies that the teacher model itself may also benefit from leveraging these intention CoTs (e.g., by appending them during inference after generating them, as described in L373–392). It would be helpful to quantify this improvement and use it as evidence for the quality and usefulness of the generated intention CoTs.

- Related to the above point, if an LLM (e.g., the teacher model) is already capable of generating intention CoTs, is Intent-FT still necessary? In other words, does the specialized training introduced by Intent-FT provide additional benefits even for models that can already generate such CoTs, or is its utility primarily for weaker models?

- Minor: The experimental results are reported from a single run. There are several instances where quotation marks are misused instead of the correct LaTeX quotes (e.g., L38, L87, L124).

**Questions:**

See the weakness.

---

> ### Author Response · Authors · 2025-11-15
>
> Dear reviewer ysq3,
>
> We are very thankful for your detailed feedback and important questions. We are glad that you found our motivation to be intiutive and our proposed method to be simple yet effective. This is indeed the goal of our work- introduce a simple fix to existing chat models while ensuring effective deterrence with minimal drawbacks.
>
> 1. **Experiment setup details**
>
> We would like to kindly point the reviewer's attention to L209-L215 and L249-258 for the experiment details. We set the epoch to 5 and conducted full parameter training.
>
> 2. **Presentation details**
>
> Thank you for your suggestions. We like to explain that the reason why we use the notations is to reserve the superscripts to differentiate between harmful $y_{\rho}^h$  and safe $y_{\rho}^s$  continuations. As you suggested, we have moved Figure 5,7,12 to the main content such that readers can easily reference the supporting figures when reading the context.
>
> 3. **Plot on Pareto Frontier**
>
> Thank you for your great suggestion! We have included a plot illustrating the trade-off between Safety (100 - ASR) and Utility in **Figure 3A (highlighted in red) in the revised manuscript**.The figure shows that when compared with stronger defensive baselines such as IA and BD-A, our Intent-FT can achieve greater safety yet with better utility retention. More importantly, we observe a significant improvement from the existing defensive baselines across all three models.
>
> 4. **More results on Gemma**
>
> As requested, we conducted additional experiments on Gemma2-9b, measuring ASR on PAIR, DI, Harmful-FT and utility scores across the three reasoning benchmarks. Based on the existing findings, we compare against more competitive baselines such as IA,BD-A and Safety-FT.
>
> |Method        | PAIR | DI | Harmful-FT |
> |:--------------|------:|------:|------:|
> | Vanilla     | 96  | 48  |  65     |
> | IA     | 59 | 0 | 52 |
> | BD-A     | 92 | 39 | 15 |
> | Safety-FT     | 90 | 16 | 62 |
> | RPO | 65 | 2 | 57 |
> | Intent-FT    | 22 | 0 | 0 |
>
> |Method        | ARC | GSM8K | GPQA |
> |:--------------|------:|------:|------:|
> | Vanilla     | 89.3| 84.8  |  32.6     |
> | IA     | 87.8 | 81.1 | 29.7 |
> | BD-A     | 82.2 | 77.9 | 30.4 |
> | Safety-FT     | 81.1 | 80.1 | 32.1 |
> | Intent-FT    | 86.2 | 83.1 | 32.2 |
>
> **We observe similar findings on Gemma2-9b as well, where Intent-FT is the best method at optimizing the defense-utility tradeoff.**
>
> **We have also include additional comparison results against another defense baseline, RPO [1] as suggested by reviewer pSrk, which further strengthens the merits of our proposed method. (Highlighted in red)**
>
> [1] Zhou, Andy, Bo Li, and Haohan Wang. "Robust prompt optimization for defending language models against jailbreaking attacks." Advances in Neural Information Processing Systems 37 (2024): 40184-40211.
>
> 5. **Are stronger models sufficient to generate intentions**
>
> This is a very good question! This comparison can actually be observed from our results in Table 1. If we consider GPT4-1 to be the  teacher model (which it is capable of doing so, given its substantially higher utility scores), then IA would be equivalent to leveraging the intention CoTs as suggested by you. However as observed, IA still incurs a high ASR as compared to Intent-FT on the weaker Llama and Qwen models.
>
> The main difference between simply prompting a LLM and intent-FT is that we use a very small set (~100) of adversarial instructions combined with the original instructions to train the model on intention-deduction. This intention-deduction teaches the model to **”decode" the adversarial instruction into the original which appears to be straightforwardly harmful, which in turn triggers the model's safety mechanism**. Thus, this is a form of adversarial training but combined with a prior reasoning step. Since the teacher model did not go through this adversarial training, the intention analysis would often break and cause the model to be fooled and comply with the instruction.
>
> Once again, we sincerely thank you for taking the time to review and providing helpful feedbacks for further improvement! We hope that the above actions have addressed your concerns. If not, what further clarification or modifications could we make to improve your score?

---

> ### Author Response · Authors · 2025-11-27
> **Kind reminders**
>
> Dear Reviewer ysq3,
>
> We sincerely thank the reviewer once again for their careful reading and constructive feedback, which have helped us significantly improve the quality and clarity of this work. We have addressed all raised concerns in detail in the rebuttal and corresponding revisions. We kindly invite the reviewer to reassess the manuscript in light of these changes, and if the responses have satisfactorily resolved the concerns, we would greatly appreciate reconsideration of the scores accordingly.

---

### Official Review · Reviewer_7yRw · 2025-10-30

**Soundness:** 3
**Presentation:** 3
**Contribution:** 3
**Rating:** 8
**Confidence:** 3

**Summary:**

This paper addresses LLMs’ safety-task performance trade-off and vulnerability to jailbreaks due to shallow alignment. It proposes INTENT-FT, an intent-aware fine-tuning method that trains LLMs to infer instruction intent first. Key results: all attack success rates (ASR) stay below 50%, over-refusal on benign prompts is reduced, and model utility is preserved across open-source (Llama, Qwen) and proprietary (GPT-4.1-mini) models.

**Strengths:**

1. Comprehensive evaluation: It tests both parametric (Harmful-FT) and non-parametric (PAIR, AA) attacks, covering open-source and proprietary models, which ensures robustness of results (e.g., Llama’s INTENT-FT reduces PAIR ASR to 19% vs Vanilla’s 88%).
2. Targeted over-refusal mitigation: Unlike baselines (e.g., Safety-FT) that increase over-refusal on XSTEST, INTENT-FT lowers refusal rates for Llama and GPT-4.1, as it trains on both harmful and benign intent deduction.
3. Intent transferability: INTENT-FT-generated intentions, when added as context to vanilla models, significantly reduce jailbreak ASR (e.g., GPT-4.1 achieves near-perfect deterrence), showing intent generalizability.

**Weaknesses:**

1. Limited white-box attack coverage: It only tests Ablation and ActAdd white-box methods; other techniques like CipherChat or Autodan are unexamined, which may underestimate real-world threats—adding these stronger attacks could improve generalizability.
2. Narrow dataset scale testing: The impact of D_I size is only tested up to 100 samples; larger D_I (e.g., 500+) or diverse datasets (e.g., industry-specific harmful prompts) are untested, leaving scalability unclear.

**Questions:**

Please refer to the weaknesses above.

---

> ### Author Response · Authors · 2025-11-12
>
> We are extremely thankful to reviewer 7yRw for the thoughtful, detailed feedback. We were glad to hear that you found our work presents a significant improvement over existing baselines and more importantly the extensive evaluation performed to support this.
>
> We would like to address your concerns below.
>
> 1. **CipherChat and Autodan**: We would like to kindly remind the reviewer that CipherChat and Autodan are not strictly white-box techniques. CipherChat is a prompt-based method, where the victim model is prompted to communicate in non-natural language. While Autodan requires access to the gradients of the victim model to optimize the suffix string, the method does not actually edit the hidden activations of the model unlike Ablation and Addact which does. More importantly, the main reason as to why we do not evaluate these methods in the main result is due to the **ineffectiveness** of these methods. We observe < 50% ASR on the Vanilla model without any defenses applied. This is much lower than all the other attacks we investigated and would not present any significant comparisons amongst the defense baselines. We kindly point the reviewer to **L234-235**, where we have discussed this detail.
>
> 2. **Low sample size**: We agree with the reviewer that it would be desirable to evaluate across a larger sample group. However, some of the attacks such as PAIR and AdaptiveAttack are computationally expensive and also incurs a high cost in API usage. More importantly, we like to point out that despite there only being 100 prompts being evaluated, each prompt is actually assessed a total of $B$ times on optimization-based attacks, where $B$ refers to the optimization budget. In PAIR, B is set to $45$, meaning to say, the total number of prompts evaluated on PAIR alone is closer to 4500. This budget is even higher in AA, where it is set to $200$. Lastly, we chose Jailbreakbench because it is a diverse dataset, spanning across $11$ malicious categories, as discussed in L131.
>
> Once again, we thank you for taking the time to review our work and providing helpful feedback! We hope that the above actions have addressed your concerns. If not, what further clarification or modifications could we make to improve your score?

---

### Official Review · Reviewer_pSrk · 2025-10-31

**Soundness:** 3
**Presentation:** 3
**Contribution:** 2
**Rating:** 2
**Confidence:** 3

**Summary:**

This paper proposes a fine-tuning framework termed Intent-FT, which aims to enhance LLM’s ability to infer the underlying intent of an instruction before generating a response. By modelling intent understanding, this approach seeks to improve robustness against both parametric and non-parametric attacks.

**Strengths:**

1. This paper proposes an intent-aware lightweight fine-tuning framework (Intent-FT) that incorporates intent inference to strengthen model’s defense against jailbreak attacks.

2. Compared to defense baselines, the experimental results show that the proposed Intent-FT consistently exhibits robust defense across different attack types and models.

**Weaknesses:**

1. The paper's core contribution is the introduction of fine-tuning for identifying the intention behind a query. While I respect the idea, the contribution is relatively incremental and provides limited further insight into defense.

2. Although the evaluation includes Adaptive Attack, some recent attack methods such as I-GCG [1] and DRL [2] are not considered. Including these would provide a more comprehensive assessment.

[1] Jia, Xiaojun, et al. "Improved Techniques for Optimization-Based Jailbreaking on Large Language Models." The Thirteenth International Conference on Learning Representations. 2025.

[2] Chen, Xuan, et al. "When llm meets drl: Advancing jailbreaking efficiency via drl-guided search." Advances in Neural Information Processing Systems 37 (2024): 26814-26845.

3. The current evaluation is limited to two open-source models. To demonstrate the general effectiveness and scalability of Intent-FT, it would be beneficial to include additional open-source models, such as Qwen-3-8B and Mistral-7B-Instruct-v0.3.

4. The related work and experimental comparison could be enhanced by including and discussing recent defense methods like adversarial tuning [3] and RPO [4].

[3] Liu, Fan, Zhao Xu, and Hao Liu. "Adversarial tuning: Defending against jailbreak attacks for llms." arXiv preprint arXiv:2406.06622 (2024).

[4] Zhou, Andy, Bo Li, and Haohan Wang. "Robust prompt optimization for defending language models against jailbreaking attacks." Advances in Neural Information Processing Systems 37 (2024): 40184-40211.

5. Fine-tuning typically involves balancing model robustness, generalization, and computational efficiency. This paper could be further improved by discussing the limitations and trade-offs associated with the proposed fine-tuning framework.

**Questions:**

Please refer to the concerns highlighted in the Weaknesses.

---

> ### Author Response · Authors · 2025-11-15
>
> Dear reviewer pSrk,
>
> Thank you for taking the time in reading and providing constructive feedbacks! We are glad that you acknowledged our work to be sufficiently robust against various attacks. This was our intention from the start, which is to perform a comprehensive evaluation to ensure our introduced defensive method can gaurantee a robust and generalizable defense.
>
> **1. Limited insight into defense**
>
> We respectfully disagree with the reviewer's comment. Most defensive works only prove the effectiveness of their method through black-box evaluations- by assessing the final output strings for signs of harmfulness. **Our work went far beyond that, by also performing white-box evaluations**, assessing worst-case scenarios where the attacker have full open access to the model weights. Furthermore, we even provided **mechanistic insights in Appenfix C.3**, where we interpret the reasons as to why our method is more desirable.
> - **Reduced Refusal Norms (Figure 9) and logit lens heatmap (Figure 11)**: Since the refusal direction represents a Harm–Harmless concept vector derived from the input space, the reduced norm indicates that training the model to explicitly reason about its intentions prevents it from immediately encoding a binary belief of whether to refuse. This interpretation is further supported by the logit-lens heatmap—showing the top-5 predicted tokens across layers—where no clear divergence is observed between harmful and harmless instructions, in contrast to the Vanilla model. This demonstrates that Intent-FT encourages the model to internalize refusal as a distributed process that emerges over later tokens, rather than as an early, hard-coded decision. Such behavior provides robustness against prefix-based jailbreaks, which exploit the shallow alignment phenomenon by influencing early-token activations.
> - **PCA comparison (Figure 12)**: The similarity between the PCA plots of both Vanilla and Intent-FT indicates that the Intent-FT model still maintains a clear separability between harmful and safe instructions.
>
> In addition, few existing works explore parametric attacks and commonly only evaluate adversarial attacks that optimize over the harmful instruction string, whereas we include both in our evaluation.
>
> **2. Comparisons with additional attacks (DRL and I-GCG)**
>
> As we discussed in L258-260, we did not implement gradient-based attacks such as GCG and Autodan due to their limited effectiveness. However in light of the reviewer's request for additional gaurantees, we implemented evaluations on Llama-8b, comparing only the effective defensive baselines due to API budget constraints. **Note that this includes the additional defense baseline as requested in #4.**
>
> |Llama        |I-GCG | DRL |
> |:--------------|------:|------:|
> | Vanilla     | 45  | 89  |
> | IA     | 3 | 34 |
> | BD-A     | 5 | 48 |
> | RPO      | 10 | 56 |
> | Intent-FT (Ours)   | **1** | **5** |
>
> As observed from the table above, **we observe similar conclusion from our initial evaluations- Intent-FT is equally robust against gradient-based attacks (I-GCG) and attacks derived from optimized attackers (DRL).**
>
> **3. Additional comparisons with other models**
>
> We kindly remind the reviewer that our evaluations do include Qwen. In light of another reviewer's request for results on an additional model, we expanded our evaluations on Gemma2-9b, which we show in the table below.
>
> |Method        | PAIR | DI | Harmful-FT |
> |:--------------|------:|------:|------:|
> | Vanilla     | 96  | 48  |  65     |
> | IA     | 59 | 0 | 52 |
> | BD-A     | 92 | 39 | 15 |
> | Safety-FT     | 90 | 16 | 62 |
> | RPO | 65 | 2 | 57 |
> | Intent-FT    | 22 | 0 | 0 |
>
> **Once again, we observe that our findings on the 3 models evaluated generalized towards the additional Gemma model**.
>
> **4. Additional comparisons with defensive baselines (RPO and Adversarial Tuning).**
>
> Since the code for adversarial tuning is unavailable, we implemented RPO instead. Our findings are **highlighted in red in the edited manuscript**, which we kindly ask the reviewer to refer to. The summarized finding is that **RPO underperforms Intent-FT and also incurs a high over-refusal rate on XSTest**.
>
> **5. Limitations and trade-offs**
>
> While Intent-FT involves parameter tuning, the overall training cost remains modest. The method requires only about $100$ or even as little as $60$ (see Fig. 5) adversarial safety examples, and the full fine-tuning can be completed within half an hour on standard A100 GPUs. The only additional requirements are access to adversarial instructions and a teacher LLM to produce reasoning traces Overall, this lightweight cost is well justified by the substantial gains in safety robustness and the marked reduction in negative side effects such as utility degradation and over-refusal.
>
> Once again, we thank you for taking the time to review our work! We hope we have addressed your concerns. If not, what further clarification could we make to improve your score?

---

> ### Author Response · Authors · 2025-11-27
> **Kind Reminders**
>
> Dear Reviewer pSrk,
>
> We sincerely thank the reviewer once again for their careful reading and constructive feedback, which have helped us significantly improve the quality and clarity of this work. We have addressed all raised concerns in detail in the rebuttal and corresponding revisions. We kindly invite the reviewer to reassess the manuscript in light of these changes, and if the responses have satisfactorily resolved the concerns, we would greatly appreciate reconsideration of the scores accordingly.

---

### Author Response · Authors · 2025-11-27
**Key Summary of Clarifications and Additional Experiments (Part 1)**

Once again, we thank the all reviewers for their time in reading and providing constructive feedbacks for further improvement. Here are the overview of the changes we made to our draft:

**1. Experiment: Additional evaluation against gradient-based (I-GCG) and RL Optimization-based (DRL) attacks (Reviewer pSrk)**.

As requested, we ran additional evaluations on I-GCG and DRL across competitive baselines on Llama-3.1 (see below)

|Llama        |I-GCG | DRL |
|:--------------|------:|------:|
| Vanilla     | 45  | 89  |
| IA     | 3 | 34 |
| BD-A     | 5 | 48 |
| RPO      | 10 | 56 |
| Intent-FT (Ours)   | **1** | **5** |

Firstly, we observed the same findings as discussed in our initail draft: gradient-based attacks such as I-GCG are uneffective against modern LLMs, even without any additional defensive frameworks. More importantly, results on I-GCG and DRL further strengthens the effectiveness of our framework (Intent-FT), where a clear advantage over the baselines can be observed.

**2. Experiment: Additional comparison with a defensive baseline RPO (Reviewer pSrk)**

We also studied another defensive method as requested: RPO and found that not only is RPO a weaker method at preventing harmful compliance,  it results in an execessive over-refusal rate on benign prompts, far beyond the previous baselines compared. See below and in the revised manuscript (in red).

|Defense | Llama | Qwen |
|:--------------|------:|------:|
| Vanilla     | 6.4  |4.0 |
| IA     | 16.4 | **1.6** |
| BD-A     | 25.6 | 15.6 |
| RPO      | 44 | 67.6 |
| Intent-FT (Ours)   | **2** | 8.4 |

**3. Experiment: Additional results on another LLM: Gemma (Reviewer pSrk and ysq3)**

We performed additional evaluation on Gemma-2-9b besides the existing ones (Llama, Qwen and GPT4.1), on selected metrics: ASR (PAIR, DI and Harmful-FT) and reasoning utility.  As expected, we find that Intent-FT is robust and generalize well across different models, where it can effectively deter adversarial attacks while minimizing deterioriation on its general capability. See below.

|Method        | PAIR | DI | Harmful-FT |
|:--------------|------:|------:|------:|
| Vanilla     | 96  | 48  |  65     |
| IA     | 59 | 0 | 52 |
| BD-A     | 92 | 39 | 15 |
| Safety-FT     | 90 | 16 | 62 |
| RPO | 65 | 2 | 57 |
| Intent-FT (Ours)    | 22 | 0 | 0 |

|Method        | ARC | GSM8K | GPQA |
|:--------------|------:|------:|------:|
| Vanilla     | 89.3| 84.8  |  32.6     |
| IA     | 87.8 | 81.1 | 29.7 |
| BD-A     | 82.2 | 77.9 | 30.4 |
| Safety-FT     | 81.1 | 80.1 | 32.1 |
| Intent-FT (Ours)    | 86.2 | 83.1 | 32.2 |

**4. Additional Figure: Pareto plot and moving figures to main content (Reviewer ysq3)**

We included a new figure (Figure 2a), depicting the tradeoff between ASR and utility across the defensive methods. The figure shows that Intent-FT optimizes this trade-off best across all 3 models.

We have also moved the important figures (4 and 6) to the main content so that readers can reference these figures while reading the related text. Both of these changes are highlighted in red in the revised manuscript.

---

> ### Author Response · Authors · 2025-11-30
> **Key Summary of Clarifications and Additional Experiments (Part 2)**
>
> Additionally, we would like to summarize the important clarifications on the concern and questions brought forward by the reviewers:
>
> **1. Limited insight into defense of safety frameworks and sufficient exploration on robustness of jailbreaks (Reviewer pSrk and 1TYP)**
>
> We respectfully disagree with this comment. As a matter of fact, we implemented an evaluation far beyond most existing works, such as the baselines compared in our work. Specifically, the evaluation involves
> -  **Defense effectiveness**: Adversarial attacks under both non-parametric (optimization/manual) and parametric setting.
> - **Defense effectiveness**: White-box attacks, access to model weights
> - **Side-effects**: Utility (reasoning) retention on both open and closed-ended format benchmarks.
> - **Side-effects**: Excessive refusal on benign instructions that include harmful words such as "kill, smash, destroy, etc"
>
> We also disagree with reviewer 1TYP's remark on there being sufficient exploration on jailbreak attacks. To the best of our knowledge, our work is the first defensive methodology work that performs a white-box attack evaluation. Because of this, we were able to establish another interesting value from our method: **Intent-FT induces a trade-off between compliance and harmful capability, thereby reducing harmful compliance.** We think that as the community improves towards an era where open-sourced models are sufficiently capable to do harm; Intent-FT is a viable training framework that can be applied on-top of the base training methodology. So that even in the event where the attacker has knowledge of how to manipulate the weights of the model for compliance, the intention training would heavily reduce the ability of the model to correctly carry out the attack.
>
> **2. Strong models alone are sufficiently robust to adversarial attacks (Reviewer 1TYP and ysq3)**
>
> If this statement is true, then GPT4.1 with IA should be competitive or better than weaker models such as Llama and Qwen trained with Intent-FT. However, our findings in Table 1 shows the opposite, where given a  sufficient budget, optimization-based attacks can still break through strong models with defensive prompting strategies such as IA. Furthermore, as seen in Figure 17, a model can produce a safe intention analysis but still comply with the request, displaying a disconnect between harmful recognition and safety compliance. We find that fine-tuning is still required to bake both of these qualities into the model in a manner that enables them to be robust.
>
> We hope our clarifications and experiments is successful at clearly up any remaining concerns on our work.

---

### Meta-Review · Area_Chair_28M2 · 2026-01-11

**Summary:**

The paper receives no post-rebuttal discussion from the reviewers despite sincere efforts of the authors to respond to reviewers' questions. The AC read every line of review comments and authors' responses. The AC found that (1) concerns of reviewer pSrk (rating 2) seems not completely addressed by the authors' repsonses (No answer to the incremental contribution argument. Not very convincing argument about limited insight. Mentioned limitation is not really limitations. Although it is encouraging that the authors include the result of I-GCG, DRL and Gemma experiments), (2) concerns of reviewer 1TYP (rating 4) seems limitedly addressed (Answer to the question of why ASR is below 50% is by quoting results reported in AutoDAN (36%) but not sure about the reasons. Other questions are answered by mentioning the results of IA -- seems reasonable). As the negative concerns still persists and the authors' response seems not convincingly addressing them, the AC recommends the authors to revise and submit to relevant venues as the paper is very promising to be reported to the community.

**Reviewer Concerns:**

Concerns by 1TYP are not well addressed.

**Reviewer Scores:**

All reviewers have no participation in the discussion.

---

### Decision · Program_Chairs · 2026-01-26

Reject